# Evidence for multiferroicity in single-layer CuCrSe$_2$

Zhenyu Sun[1,2,3,10], Yueqi Su[4,5,6,10], Aomiao Zhi[1,3,10], Zhicheng Gao[1,3], Xu Han[7], Kang Wu[1,3], Lihong Bao[1,3], Yuan Huang[7], Youguo Shi[1,3], Xuedong Bai[1,3,8], Peng Cheng[1,3], Lan Chen[1,3,8]✉, Kehui Wu[1,3,8,9], Xuezeng Tian[1,3]✉, Changzheng Wu[4,5,6]✉ & Baojie Feng[1,3,8,9]✉

Multiferroic materials, which simultaneously exhibit ferroelectricity and magnetism, have attracted substantial attention due to their fascinating physical properties and potential technological applications. With the trends towards device miniaturization, there is an increasing demand for the persistence of multiferroicity in single-layer materials at elevated temperatures. Here, we report high-temperature multiferroicity in single-layer CuCrSe$_2$, which hosts room-temperature ferroelectricity and 120 K ferromagnetism. Notably, the ferromagnetic coupling in single-layer CuCrSe$_2$ is enhanced by the ferroelectricity-induced orbital shift of Cr atoms, which is distinct from both types I and II multiferroicity. These findings are supported by a combination of second-harmonic generation, piezo-response force microscopy, scanning transmission electron microscopy, magnetic, and Hall measurements. Our research provides not only an exemplary platform for delving into intrinsic magnetoelectric interactions at the single-layer limit but also sheds light on potential development of electronic and spintronic devices utilizing two-dimensional multiferroics.

The exploration of multiferroic materials, which simultaneously exhibit ferroelectric and ferromagnetic orders has been at the forefront of condensed matter physics due to their fascinating fundamental properties and promising applications in spintronics and information technology[1-4]. In these materials, the interplay between magnetic and electric orders gives rise to a plethora of cross-correlated phenomena[1-8], enabling the manipulation of the magnetic states by electric fields, and vice versa. Over the past decades, the trend towards device miniaturization has stimulated great research interest in two-dimensional (2D) materials[9-14]. Achieving 2D multiferroic materials[15] is highly desirable for practical device applications. However, as thickness diminishes, both long-range ferromagnetic and ferroelectric orders tend to be suppressed by mechanisms such as thermal fluctuations and depolarization fields. The realization of either 2D ferromagnetic[16-18] or ferroelectric[19-27] materials at the single-layer (1L) limit has been reported relatively recently, and these often come with notably reduced transition temperatures.

Despite recent advancements in 2D ferromagnetic and ferroelectric materials, achieving 2D multiferroicity that integrates both ferromagnetic and ferroelectric orders remains a formidable

[1]Institute of Physics, Chinese Academy of Sciences, Beijing 100190, China. [2]Department of Chemistry, Brown University, Providence, RI 02912, USA. [3]School of Physical Sciences, University of Chinese Academy of Sciences, Beijing 100049, China. [4]School of Chemistry and Materials Sciences, University of Science and Technology of China, Hefei 230026, China. [5]CAS Center for Excellence in Nanoscience, and CAS Key Laboratory of Mechanical Behavior and Design of Materials, Hefei 230026, China. [6]Collaborative Innovation Center of Chemistry for Energy Materials (iChEM), Hefei 230026, China. [7]Advanced Research Institute of Multidisciplinary Science, Beijing Institute of Technology, Beijing 100081, China. [8]Songshan Lake Materials Laboratory, Dongguan, Guangdong 523808, China. [9]Interdisciplinary Institute of Light-Element Quantum Materials and Research Center for Light-Element Advanced Materials, Peking University, Beijing 100871, China. [10]These authors contributed equally: Zhenyu Sun, Yueqi Su, Aomiao Zhi. ✉e-mail: lchen@iphy.ac.cn; tianxuezeng@iphy.ac.cn; czwu@ustc.edu.cn; bjfeng@iphy.ac.cn

challenge. Even in three-dimensional (3D) materials, multiferroicity is rare, as the origins of magnetism and ferroelectricity are often mutually exclusive[2]. There are a few experimental instances of 2D multiferroicity in few-layer materials, including $p$-doped SnSe[28] and $\varepsilon$-Fe$_2$O$_3$[29]. However, these materials, typically thicker than 3 nm, exhibit significantly suppressed multiferroic order as their thickness decreases. Recently, Song *et al.* reported a type-II multiferroic ground state in single-layer NiI$_2$, identified through various optical techniques[30]. Nonetheless, the low transition temperature ($T = 21$ K) and poor air stability of NiI$_2$ pose serious impediments to its practical application. Furthermore, the all-optical characterization approach used to determine multiferroicity in single-layer NiI$_2$ has been met with skepticism[31], and more direct measurements such as piezo-response force microscopy (PFM) and magnetic measurements are greatly awaited. Another significant advancement in the pursuit of single-layer multiferroicity is the fabrication of one-unit-cell-thick multiferroic Cr$_2$S$_3$[32]; however, its 2D ferroelectric properties are attributed to interfacial modulation. Hence, the quest for a single material that exhibits intrinsic 2D multiferroicity remains a formidable challenge.

In this work, we focus on the non-van der Waals material CuCrSe$_2$, which has garnered significant interest due to its excellent thermo-electric properties[33–35]. We provide compelling evidence for the multiferroicity in single-layer CuCrSe$_2$. This material demonstrates out-of-plane polarization at room temperature, with a large coercive field, as substantiated by various techniques including second harmonic generation (SHG), PFM, and cross-section high-resolution scanning transmission electron microscopy (STEM). Ferromagnetism with a Curie temperature of approximately 120 K was confirmed by magnetic measurements. Further analysis reveals that the ferroelectricity in CuCrSe$_2$ originates from the displacements of interlayer Cu atoms, leading to an orbital shift in Cr atoms of the CrSe$_2$ layers. This orbital rearrangement of Cr atoms plays a critical role in stabilizing the ferromagnetic coupling.

## Results

### Crystal structure and chemical exfoliation of single-layer CuCrSe$_2$

CuCrSe$_2$ crystallizes in the R3m space group, with alternating layers of CrSe$_2$ and Cu atoms stacked along the $c$-axis[36], as denoted by the labels A, B, and C in Fig. 1a. This structure was confirmed by the x-ray diffraction measurements (see Supplementary Fig. 1). Within each CrSe$_2$ layer, the Cr$^{3+}$ cation is octahedrally coordinated by six Se$^{2-}$ ions, as illustrated in Fig. 1a. Although the unit cell of bulk CuCrSe$_2$ comprises three Cu sheets and three CrSe$_2$ layers, we define single-layer (1 L) CuCrSe$_2$ as consisting of two CrSe$_2$ layers and one Cu layer, which are strongly covalently bonded to each other, as illustrated in Fig. 1b. The addition of one more CuCrSe$_2$ layer includes one more CrSe$_2$ layer and one Cu layer, respectively. Although the chemical formula should more accurately be denoted as CuCr$_2$Se$_4$, we continue to use "1L CuCrSe$_2$" for simplicity and to avoid confusion, given that CuCrSe$_2$ is the chemical formula of the bulk crystal. Notably, the vertical distances between Cu and Se atoms differ ($d_1 \neq d_2$), leading to a vertical (downward) electric polarization, $P_Z$. This vertical polarization induces shifts in the original Cr-d orbitals in the Cr$^{3+}$ configuration, primarily $e_g$ and $t_{2g}$ resulting from the octahedral crystal field splitting, as depicted in Fig. 1c. Specifically, the $t_{2g}$ orbitals of one Cr$^{3+}$ ion experience a downward energy shift, while the $e_g$ orbitals of another undergo an upward shift. Consequently, the energy difference $e_g$(Cr$_1$)-$t_{2g}$(Cr$_2$) diminishes, thereby enhancing the ferromagnetic coupling within the material[37]. In this context, a single layer of CuCrSe$_2$ is intrinsically multiferroic, as its electric polarization is intimately linked to its magnetic properties. Additionally, single-layer CuCrSe$_2$ exhibits significant hole doping, which effectively increases the carrier density at the Fermi level, favoring both intralayer and interlayer ferromagnetic couplings.

Through redox-controlled chemical exfoliation[38], we selectively removed partial Cu$^+$ ions at the interface while tetraoctylammonium intercalation led to a marked increase in interlayer spacing. This

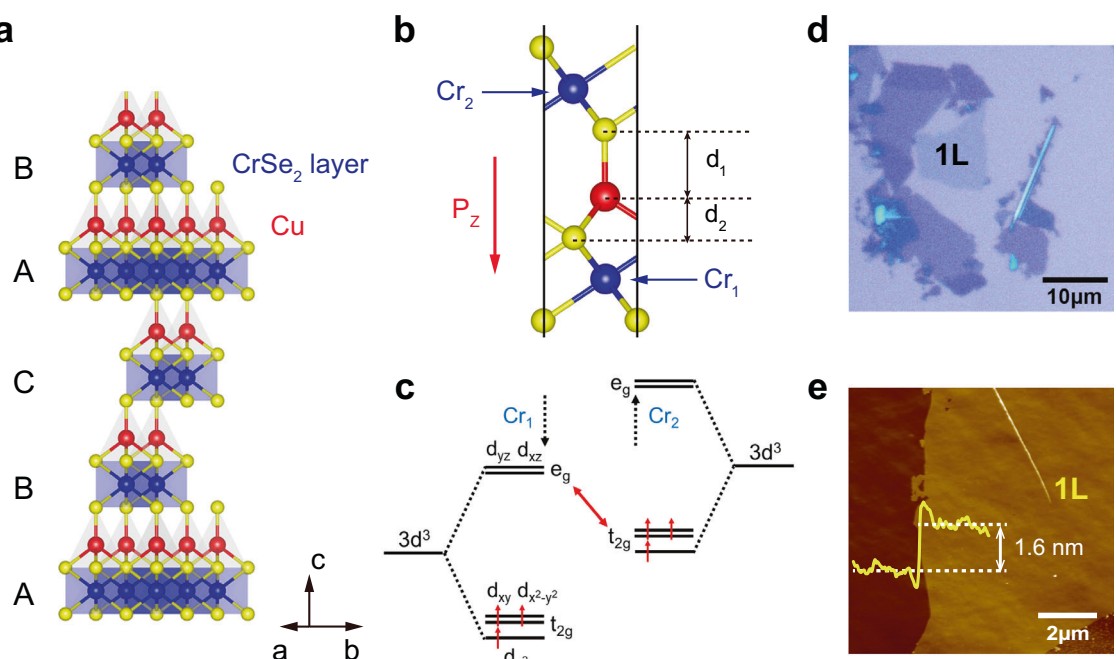

**Fig. 1 | Crystal structure and chemical exfoliation of single-layer CuCrSe$_2$. a** Side view of the atomic structure of CuCrSe$_2$. Red, blue, and green balls indicate Cu, Cr, and Se atoms, respectively. **b** Crystal structure of single-layer CuCrSe$_2$, highlighting two distinct Cr atoms, labeled as "Cr$_1$" and "Cr$_2$". **c** Analysis of Cr orbital energy of single-layer CuCrSe$_2$, illustrating the vertical polarization-induced orbital energy shift in opposing directions. **d** Representative optical image of chemically exfoliated CuCrSe$_2$ flakes, with the single-layer region identified. Scale bar: 10 μm. **e** AFM image of the single-layer region, scale bar: 2 μm. Inset shows the height profile across the sample step.

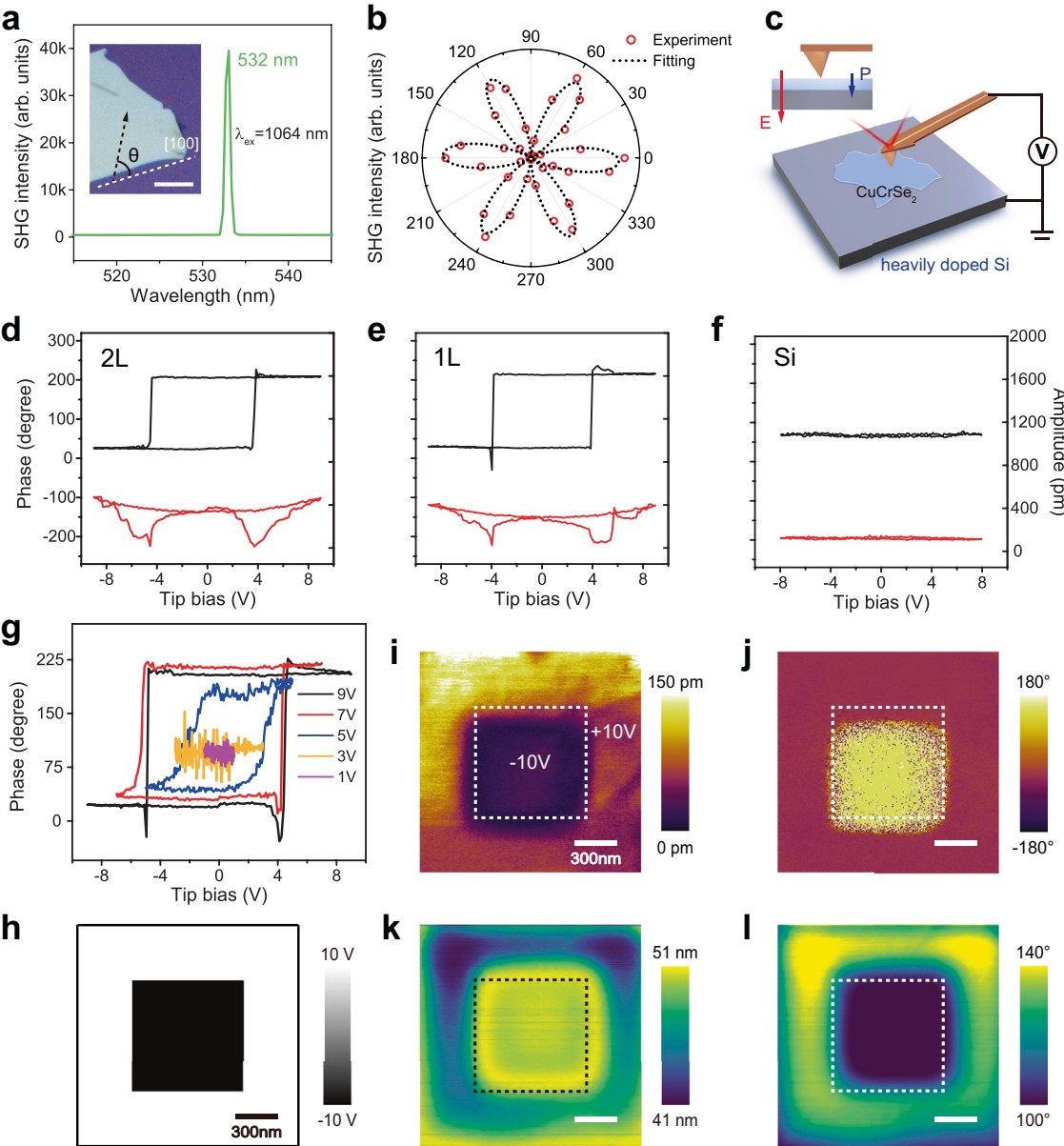

**Fig. 2 | Out-of-plane ferroelectricity in ultrathin CuCrSe₂. a** SHG spectrum of a CuCrSe₂ flake, excited with a 1064 nm picosecond laser and $\theta = 0°$. Inset: The corresponding optical image showcases the sample, with the angle $\theta$ defined as the relative angle between the [100] axis (indicated by a white dashed line) and the direction of laser polarization. Scale bar: 20 μm. **b** Polarization angle-dependent SHG intensity measured in the parallel configuration. **c** Schematic diagram of the PFM measurement setup. **d**–**f** Off-field PFM phase and amplitude loops for bilayer (2L) CuCrSe₂, single-layer (1L) CuCrSe₂, and a bare silicon substrate, respectively. **g** Tip bias-dependent ferroelectric hysteresis loop acquired on single-layer CuCrSe₂. **h** "Box-in-box" pattern used for PFM domain writing. Scale bar: 300 nm. **i, j** PFM amplitude and phase images after opposite DC bias writing. Scale bar: 300 nm. **k, l** EFM amplitude and phase images on the same area as (**i, j**). Scale bar: 300 nm.

process ultimately facilitated the fabrication of ultrathin 2D CuCrSe₂ nanosheets from high-quality single crystals (for further details, see Methods). Figure 1d displays a typical optical image of such exfoliated CuCrSe₂ nanoflakes, where the monolayer region can be distinguished by its minimal optical contrast. AFM measurements were conducted on this 1L flake, as depicted in Fig. 1e, revealing a pristine surface with a step height of approximately 1.6 nm relative to the substrate. This measured thickness corresponds to 1L CuCrSe₂, which is slightly larger than the theoretical value of ~1.3 nm. Similar overestimations have been reported for many other 2D materials[39,40]. Subsequent Raman spectra were obtained for the exfoliated CuCrSe₂ nanoflakes (see Supplementary Fig. 2). Analogous to CuCrS₂[41], two distinct Raman modes, $E$ (-146.2 cm⁻¹) and $A_1$ (-237.7 cm⁻¹), emerge, corresponding to in-plane and out-of-plane vibrational modes, respectively. It is

noteworthy that these Raman modes are nearly undetectable in the single-layer limit.

## Out-of-plane ferroelectricity in ultrathin CuCrSe₂

As previously noted, the occupation of Cu⁺ ions in CuCrSe₂ induces a spontaneous electric polarization. This structural asymmetry is detectable through nonlinear optical techniques such as SHG[42,43]. For our experiments, we employed a 1064 nm picosecond laser as the excitation source based on the backscattering geometry and observed a robust emission peak centered at 532 nm, as shown in in Fig. 2a. This wavelength is precisely half of the incident wavelength, thereby underscoring the symmetry-breaking nature of the CuCrSe₂ lattice. Subsequent polarization-dependent SHG measurements in the parallel-polarized configuration yielded a six-fold symmetry pattern, as

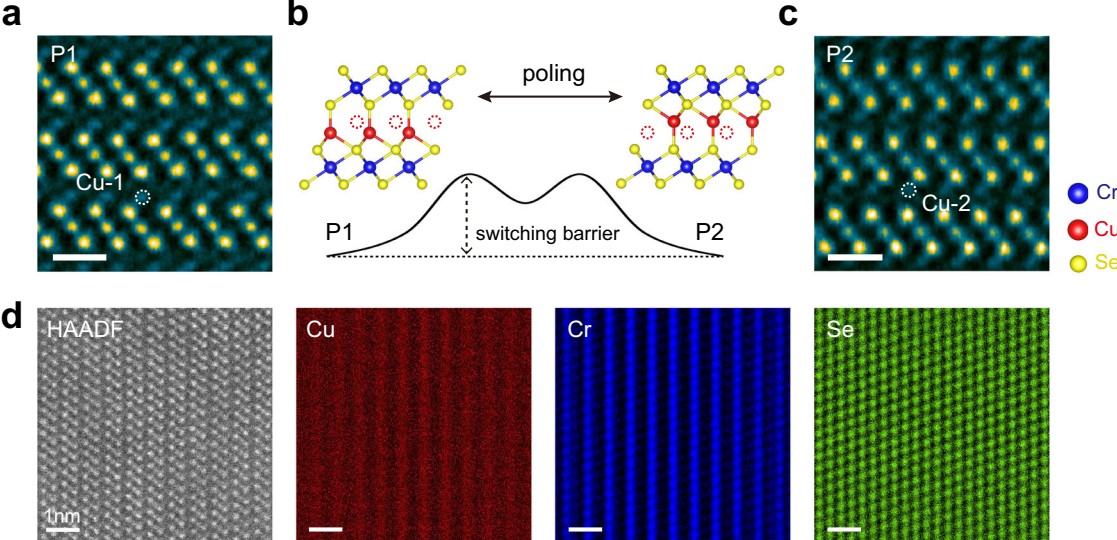

**Fig. 3 | STEM measurement of CuCrSe₂. a, c** Cross-section high-resolution STEM images of the two ferroelectric CuCrSe₂ phases along the [100] zone axis, P1 and P2, where the dashed circles highlight two different Cu occupation sites. Scale bar: 0.5 nm. **b** Illustration of the two polarization states in single-layer CuCrSe₂, along with the ferroelectric switching pathway between these states. **d** Representative high-angle annular dark-field (HAADF) STEM image of CuCrSe₂ and corresponding energy-dispersive spectroscopy (EDS) maps. Scale bar: 1 nm.

shown in Fig. 2b. This result is consistent with the crystalline structure of CuCrSe₂ and fits well with the cosine square formula. In summary, the SHG response of CuCrSe₂ nanoflakes confirms its intrinsic non-centrosymmetric nature, fulfilling the basic criterion for ferroelectricity.

To validate the presence of room-temperature out-of-plane ferroelectricity in ultrathin CuCrSe₂ flakes, we conducted a comprehensive set of PFM measurements. The experimental schematic for our PFM setup is illustrated in Fig. 2c, where a silicon chip served as a conductive substrate to mitigate charge accumulation. Typically, applying a minor voltage to the AFM tip induces uniform dipole moment orientation in ferroelectric materials under the influence of a vertical electric field[44]. We recorded off-field phase and amplitude loops for bilayer CuCrSe₂, single-layer CuCrSe₂, and the silicon substrate, as depicted in Fig. 2d–f. For both the bilayer and single-layer CuCrSe₂ flakes, we observed a distinct hysteresis in the phase loop accompanied by a 180° phase switch, along with a well-defined butterfly-shaped amplitude evolution. In contrast, no piezo-response signal was detected for the bare silicon substrate. The local PFM switching spectra corroborate the out-of-plane ferroelectricity inherent in this system. Moreover, the coercive voltage is found to lie in the range of 4–5 V, which exceeds that of many conventional 2D ferroelectrics like In₂Se₃ (1.5 V)[45], SnS (2.0 V)[46], and MoS₂/WS₂ heterobilayer (3.0 V)[47]. This suggests that CuCrSe₂ possesses a more stable ferroelectric state characterized by a higher energy barrier separating the two distinct polarization states. In addition, compared with conventional layered 2D ferroelectrics, the emergence of room-temperature ferroelectricity in non van der Waals CuCrSe₂ broadens the spectrum of ferroelectric materials.

To rule out alternative mechanisms that could generate ferroelectric artifacts, such as Cu⁺ ion migration, we analyzed the direct current voltage ($V_{DC}$) dependent phase curve[48]. The results are summarized in Fig. 2g. In an intrinsic ferroelectric material, when the maximum sweeping $V_{DC}$ surpasses the coercive voltage, the switching threshold voltage remains largely consistent, exhibiting only minor variations. Conversely, when the maximum $V_{DC}$ falls below the coercive voltage, polarization switching is inhibited, and the PFM loops vanish. This behavior precisely aligns with our observations for 1L CuCrSe₂, as shown in Fig. 2g. In contrast to ion-mediated piezo-responses observed in materials like TiO₂ thin films[49], CuCrSe₂ exhibits structure-

mediated spontaneous polarization. In essence, this confirms the material's robust 2D ferroelectricity, surviving to the single-layer limit.

We corroborate the ferroelectric properties via PFM domain writing and reading experiments. For a 1L CuCrSe₂ nanoflake on a silicon substrate, we selected a flat region (Fig. 1e) and poled the area with opposite biases of +10 V and −10 V, in accordance with the pattern depicted in Fig. 2h. After domain writing, a distinct "box-in-box" pattern emerges in both the PFM amplitude (Fig. 2i) and phase images (Fig. 2j), revealing a marked 180° phase contrast between the two domains (as further verified in Supplementary Fig. 3). Notably, this domain pattern remains clearly identifiable even after one day and one week (see Supplementary Fig. 4), substantiating the non-volatile nature of the material's ferroelectricity. Furthermore, the antiparallel charge polarization manifests as a discernible contrast in both the electrostatic force microscopy (EFM) amplitude (Fig. 2k) and phase images (Fig. 2l).

## STEM measurements

Having demonstrated the robust out-of-plane ferroelectricity in CuCrSe₂, we further elucidate the underlying physical origins through STEM investigations. As suggested by prior theoretical calculations[37], both bulk and single-layer CuCrSe₂ possess two stable ferroelectric configurations: P1 and P2. These configurations can be switched between each other through electrical field poling, as depicted in Fig. 3b. The principal distinction between the P1 and P2 polarization states lies in the occupancy of interlayer Cu atom sites, which are designated as "Cu-1" and "Cu-2", respectively[50]. Intriguingly, we are able to capture both tetrahedron sites using cross-section high-resolution STEM in different regions of the CuCrSe₂ sample along the [100] zone axis, as illustrated in Fig. 3a, c. Here the sample shows high chemical homogeneity, as evidenced by the energy-dispersive spectroscopy (EDS) mapping in Fig. 3d, clearly revealing the Cu, Cr and Se element distributions. Each of these asymmetric configurations gives rise to a spontaneous polarization. Thus, the ferroelectric nature of CuCrSe₂ can be comprehensively understood via the traditional double-well potential model[51]. This model also explains the relatively high energy barrier for ferroelectric switching (Fig. 3b), which allows the ferroelectric state to be stable at room temperature. When Cu atoms transition from the "Cu-1" site to the adjacent "Cu-2" site, the charge center has vertical displacements without a corresponding

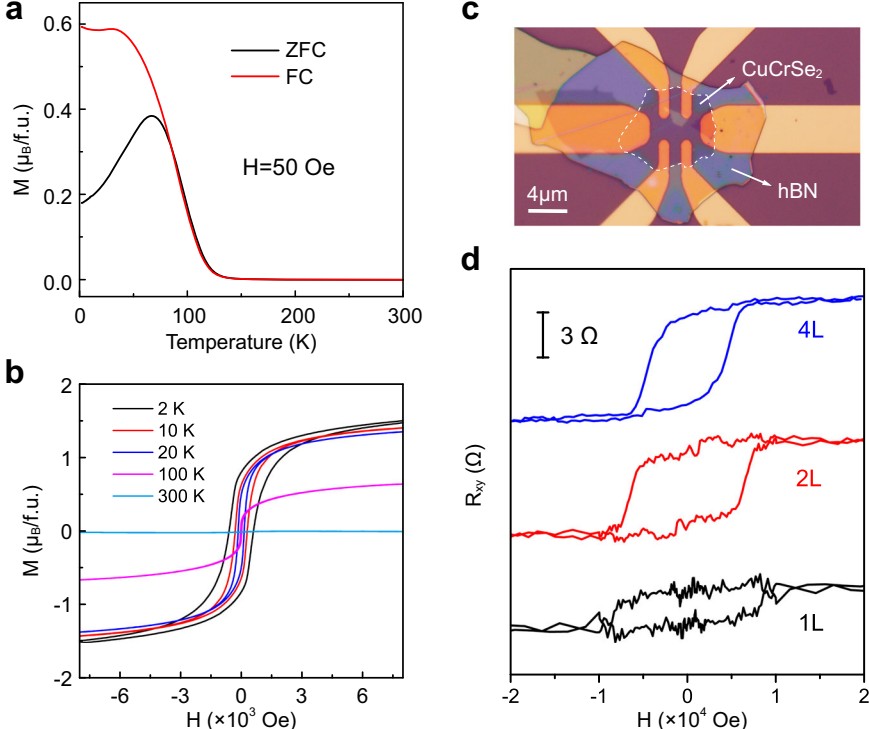

**Fig. 4 | Magnetic properties of CuCrSe₂ nanosheets. a** M-T curves for the nanosheets under a parallel magnetic field of 50 Oe. **b** M-H magnetic hysteresis loops of the nanosheets at various temperatures. **c** Optical image of the CuCrSe₂ device used for Hall measurements. Scale bar: 4 μm. **d** Out-of-plane magnetic field dependent Hall resistance for 1L, 2L, and 4L CuCrSe₂ flakes. The data were acquired at 2 K.

movement in the CrSe₂ layers. Therefore, CuCrSe₂ exhibits the out-of-plane ferroelectric switching behavior.

## Ferromagnetism in CuCrSe₂ nanosheets

In addition to exhibiting intrinsic ferroelectric properties, ultrathin CuCrSe₂ nanosheets also manifest spontaneous ferromagnetism. Figure 4a displays temperature-dependent zero-field-cooled (ZFC) and field-cooled (FC) magnetization measurements. The data indicate a pronounced increase in magnetization below 120 K under a modest external magnetic field ($H = 50$ Oe), signifying a phase transition from a paramagnetic to ferromagnetic state with a Curie temperature ($T_c$) of approximately 120 K. In the ZFC magnetization curve, a noticeable cusp is evident after its divergence from the FC curve. This feature can be attributed to the fact that the applied external magnetic field is less than the material's coercive field. Meanwhile, such a phenomenon might be ascribed to the residual antiferromagnetic coupling in the sample. Additionally, field-dependent magnetization measurements (Fig. 4b) for CuCrSe₂ films reveal a distinct magnetic hysteresis loop below the $T_c$. At a temperature of 2 K, the coercive field is approximately 600 Oe, indicative of a robust ferromagnetic ordering.

We further performed Hall measurements on ultrathin CuCrSe₂ flakes to validate its intrinsic ferromagnetism. As displayed in Fig. 4d, the single-layer sample exhibits anomalous Hall effect (AHE) with a large hysteresis window, indicating that the magnetic order can persist down to monolayer with a strong in-plane magnetic anisotropy[52]. For bilayer and quadrilayer CuCrSe₂, the AHE is still observed but with a slightly decreased coercive field. We noticed that only linear Hall effect exists in the trilayer (see Supplementary Fig. 8), which indicates an odd-even-layer-dependent ferromagnetism in CuCrSe₂.

Although bulk CuCrSe₂ exhibits an antiferromagnetic ground state owing to interlayer antiferromagnetic coupling[53], single-layer CuCrSe₂ favors a ferromagnetic state. As elucidated in Fig. 1b, c, the Cu⁺ ions contribute not only to vertical polarization but also induces an

orbital shift in Cr atoms. This orbital shift effectively augments the ferromagnetic coupling between the adjacent Cr layers of CrSe₂ in single-layer CuCrSe₂ structure. Consequently, this leads to the emergence of a naturally occurring multiferroic state in this material.

## Discussion

In summary, we have verified the presence of intrinsic 2D multiferroicity in single-layer CuCrSe₂ using a comprehensive suite of experimental techniques, including SHG, PFM, STEM, and magnetic measurements. Importantly, the transition temperatures for both ferroelectricity and ferromagnetism are notably high, exceeding room temperature and 120 K, respectively. These findings represent a seminal advancement in achieving intrinsic 2D multiferroicity in a single-layer material at comparatively elevated temperatures, thus paving the way for the potential development of advanced multi-terminal spintronic chips and magnetoelectric devices.

## Methods

### Growth of CuCrSe₂ single crystals

Single crystals of CuCrSe₂ were synthesized using a chemical vapor transport (CVT) method starting from polycrystalline powders. Polycrystalline CuCrSe₂ was prepared by a solid-state reaction in which a stoichiometric mixture of high purity Cu, Cr, and Se powders was sealed in an evacuated quartz tube and sintered at 1050 °C for 40 h. The furnace was then heated to 1200 °C and kept for several minutes, followed with rapid quenching to obtain pure phase of CuCrSe₂. The polycrystalline CuCrSe₂ was then grinded into fine powder and sealed in an evacuated quartz tube together with a small amount of CrCl₃ as a transport agent. The tube was heated on a two-zone tube furnace in a temperature gradient of 1050 °C at the hot end and 800 °C at the cold end for 200 hours. Rapid quenching with cold water was also employed and single crystals of CuCrSe₂ were obtained.

## Preparation of CuCrSe$_2$ nanosheets

CuCrSe$_2$ nanosheets were exfoliated from bulk CuCrSe$_2$ single crystals using a redox-controlled electrochemical exfoliation method. Specifically, a CuCrSe$_2$ crystal was anchored between two titanium plates, serving as the cathode, while a platinum electrode was employed as the anode. The electrolyte for the process was a 0.1 M solution of tetra-octylammonium (TOA$^+$) bromide in acetonitrile. A voltage of 5 V was applied for 1 hour during the electrochemical intercalation process. Subsequently, the intercalated CuCrSe$_2$ crystal underwent multiple washes with dimethyl formamide (DMF) and was manually shaken in DMF to exfoliate into nanosheets. The resulting CuCrSe$_2$ dispersion was centrifuged at 112 × g for 3 minutes to separate out the unexfoliated crystals. Thin flakes with different thicknesses were selected by atomic force microscopy measurements, as shown in Supplementary Fig. 7.

## Raman characterization

Raman spectroscopy measurements were performed using a confocal optical system (LabRAM HR Evolution) equipped with a 532 nm laser for excitation, conducted at room temperature. To ensure accuracy, the phonon mode of the silicon substrate (520.7 cm$^{-1}$) was employed for calibration of each acquired spectrum.

## SHG characterization

Single SHG spectrum and polarization-resolved SHG analyses were conducted on a WITec alpha 300RA Raman microscope. These measurements utilized a 1064 nm ultrafast laser with a pulse width of approximately 15 picoseconds and a repetition rate of 80 MHz for excitation. A half-wave plate, integrated into the common optical path for both incident laser and collected light, facilitated these characterizations. During polarization-resolved SHG measurements, the half-wave plate was rotated in a clockwise direction, and the analyzer was aligned parallel to the initial polarization of the laser. SHG signals were subsequently detected using a UHTS spectrometer system, coupled with a CCD detector for spectral analysis.

## High-resolution STEM characterization

The samples for STEM measurements were fabricated through focused ion beam (FIB) milling. High-angle annular dark-field (HAADF) imaging was conducted using a JEOL-ARM 300 F microscope, operated at an accelerating voltage of 300 kV. The convergence semi-angle was 22 mrad, and the collection angle ranged from 64 to 180 mrad.

## Scanning probe microscopy measurement

Atomic force microscopy (AFM), piezoresponse force microscopy (PFM), and electrostatic force microscopy (EFM) measurements were conducted using a commercial scanning probe microscope (Asylum Research Cypher S, Oxford Instruments) at room temperature. Heavily doped silicon substrates were utilized to prevent charge accumulation. PFM local switching spectroscopy was performed in dual AC resonance tracking (DART) mode, with the superimposition of an AC signal over a series of DC triangular, sawtooth waveform voltages (refer to Supplementary Fig. 9 for additional details). The PFM phase and amplitude loops were recorded when the tip voltage was zero.

## Magnetic measurement

The magnetic measurements were conducted using a SQUID magnetometer (MPMS-3, Quantum Design). The film of CuCrSe$_2$ nanosheets for magnetic measurements was prepared by filtering the CuCrSe$_2$ dispersion over on a polyvinylidene fluoride membrane. The film was subsequently kept at 80 °C for 1 hour in a vacuum oven to remove any residual DMF.

## Hall device fabrication and transport characterization

Chemically exfoliated CuCrSe$_2$ nanoflake was transferred onto the pre-patterned Au/Ti electrodes, followed by a hBN flake transfer on top of it to protect the surface. The Hall measurements were conducted using a 9 T-physical property measurement system (PPMS, Quantum Design, DynaCool).

## Reporting summary

Further information on research design is available in the Nature Portfolio Reporting Summary linked to this article.

## Data availability

The data that support the findings of this paper are available from the corresponding authors upon request.

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

## Acknowledgements

This work was supported by the Beijing Natural Science Foundation (Grant no. JQ23001), the Ministry of Science and Technology of China (Grant nos. 2018YFE0202700 and 2021YFA1400502), the National Natural Science Foundation of China (Grants nos. 12374197, 11825405, T2325028, 12134019, and 12174432), the International Partnership Program of Chinese Academy of Sciences (Grant no. 112111KYSB20200012), the Strategic Priority Research Program of Chinese Academy of Sciences (Grants no. XDB33030100), and the CAS Project for Young Scientists in Basic Research (Grant no. YSBR-047). The authors would like to thank P.C. in the Materials Growth and Characterization Center of Songshan Lake Materials Laboratory for the Hall measurement.

## Author contributions

B.F. and Z.S. conceived the project. Y.S. and C.W. prepared the $CuCrSe_2$ nanosheets. Z.S., Y.S., and Z.G. performed the AFM, Raman, PFM, EFM, and magnetic measurements. Z.S., X.H., Kang Wu, Y. H., and L.B. acquired the SHG data. A.Z., X.B., and X.T. performed the FIB process and high-resolution STEM measurements. Z.S. and Y.S. prepared the Hall devices and performed the transport measurements. Z.S., P.C., Kehui Wu, L.C., X.T., C.W., and B.F. analyzed the experimental data and discuss. Z.S. and B.F. wrote the manuscript with comments from all authors.

## Competing interests

The authors declare no competing interests.
