## [Peer Review File · Nature Communications]

REVIEWER COMMENTS

Reviewer #1 (Remarks to the Author):

The manuscript by Sun et al. presents a significant discovery of coexisting high-temperature ferroelectricity and ferromagnetism in a single-layer, non-van der Waals material, CuCrSe₂. Employing a comprehensive suite of experimental techniques such as SHG, PFM, and STEM, the authors discovered ferroelectricity induced by Cu⁺ displacement. Additionally, they confirmed the coexistence of ferromagnetism through magnetic measurements and Hall effect experiments. The manuscript is well-structured, and the presented data are robust.

I am particularly impressed by the high transition temperature, remarkable chemical stability, and the novel mechanism of multiferroicity in CuCrSe₂, where ferromagnetism is augmented by ferroelectric polarization. Given the growing interest in intrinsic 2D multiferroics, this manuscript addresses a highly relevant topic. Therefore, I recommend its publication in Nature Communications. However, before proceeding to publication, several issues should be addressed to further improve the manuscript:

1. The manuscript labels the atomic structure in Figure 1a with “A, B, C” but fails to provide a corresponding description in the main text.
2. The authors should compare the ferroelectricity in CuCrSe₂ from that observed in other 2D ferroelectrics, such as WTe₂, In₂Se₃, and CuInP₂S₆, to highlight the unique aspects of their discovery.
3. Figures 3a and 3c lack scale bars, an essential detail for interpreting the images accurately.
4. In discussing in-plane ferroelectricity in monolayer CuCrSe₂, the manuscript references hysteresis in a two-terminal memristor device. To demonstrate more effectively high- and low-resistance switching, it would be beneficial if the authors could present I-V curves obtained under varying poling voltages.
5. References to “Extended Data Fig. X” within the main text should be corrected to “Supplementary Fig. X” to ensure consistency and clarity in document navigation.
6. Figure 4a showcases temperature-dependent magnetization measurements of CuCrSe₂ nanosheets, which primarily consist of ultrathin layers of CuCrSe₂. To provide a comprehensive understanding of the material’s ferromagnetic properties, it is advisable for the authors to perform magnetization measurements on bulk CuCrSe₂.
7. A recent study published in Nature Communications (Vol. 15, p. 721, 2024) on a one-unit-cell thick, interfacially modulated multiferroic Cr₂S₃ should be discussed. Although it explores a different mechanism—relying on interfacial interactions for multiferroicity, unlike the intrinsic high-temperature multiferroicity reported here—it is pertinent for contextualizing the current work within the broader research landscape.

Reviewer #2 (Remarks to the Author):

This manuscript reports the multiferroic behavior of thin flake samples of CuCrSe₂ obtained by redox-controlled chemical exfoliation. Two-dimensional materials are one of the hottest topics in the condensed matter physics nowadays, and the development of multiferroic two-dimensional materials will attract many readers. Breaking the inversion symmetry and ferromagnetism are clearly demonstrated in the manuscript. On the other hand, I am not convinced that all the experimental data are correctly interpreted. I also wonder if the information about the methodology is well provided. I hence have several questions and comments. I may recommend the publication of this work after the authors address all the following issues.

Question 1. Is the chemical formula of the film really CuCrSe₂?

The authors first say, "The single-layer CuCrSe₂ consists of two CrSe₂ layers and one Cu layer", which literally means the composition of Cu₁Cr₂Se₄. If this is the case, I recommend that the authors should refer to the material as CuCr₂Se₄. Then, the Cu ion likely behaves as a divalent ion of $S=1/2$. This kind of valence change may happen since the samples were obtained by redox-controlled chemical exfoliation. The change in Cu valence can provide an enormous impact on the magnetism of the material.

Question 2: Is the sample really of single layer?

According to the previous report on the crystal structure of bulk CuCrSe₂, the lattice parameter c in the hexagonal setting is approximately 1.94 nm (Gagor et al., *Materials Chemistry and Physics* 146, 283 (2014)). Since the hexagonal unit cell contains three layers, the thickness of single layer is estimated to be 0.6 or 0.7 nm. The obtained step of 1.6 nm is much larger than the thickness.

Question 3: How was the 2L and 4L samples fabricated and characterized?

There is no description about the fabrication and characterization of the bilayer sample used in the PFM measurement and 2L and 4L samples used in the Hall resistivity measurement.

Question 4: In what configuration was the second harmonic generation measured?

Fig. 2a just shows that the second harmonic is generated. However, the result does not tell the presence of electric polarization because of the lacking in the information on the experimental configuration. If the incident direction of the excitation laser was normal to the sheet, Fig. 2b shows the xxx, xyy, xxy, and yyy components of the second harmonic generation tensor. The azimuthal angle dependence shows threefold symmetry and inversion breaking but not the out-of-plane polarization. For example, P-6 system with no polarization may also show a similar result. In addition, the authors should explicitly show the angles corresponding to the a and b axes of the crystal.

Question 5:

How thick is the nanosheet used for the magnetization measurement demonstrated in Fig. 4?

Comment 1:

The authors claim that CuCrSe₂ also hosts the in-plane polarization, based on the double-

well model shown in Fig. 3b. However, both of P1 and P2 states have the symmetry of $R3m$, at least in the bulk case. In other words, (110) , $(2 -1 0)$, and $(-1 2 0)$ planes are reflection mirrors, which prohibits the in-plane polarization. Cu ion can move to one of the three adjacent unoccupied tetrahedral sites. Even if all the Cu ions move in the same direction at the same time, this should be regarded as ionic conduction, which has intensively been discussed since 1980's.

Comment 2:

The order-disorder type structural phase transition in CuCrSe_2 was reported in 2014 by Gagor and collaborators (Materials Chemistry and Physics 146, 283 (2014)). The paper also reported the ordering of the tetrahedral site occupation was essential for the transition. I believe that this paper has to be cited.

Comment 3:

Y. Zhou et al. very recently reported the ferroelectric thin film of CuCrS_2 (Nano Letters 24, 2118, (2024)), which should be added as a reference.

Comment 4:

Can the authors explain why the Hall resistance is smaller in a thinner flake? It is counterintuitive.

Reviewer #3 (Remarks to the Author):

2D multiferroic materials have garnered broad interests attributed to their magnetoelectric properties and multifunctional applications. The simultaneous coexistence of robust magnetism/ferroelectricity in ultrathin-layer CuCrSe_2 was previously predicted via first-principles calculations by Liu et al in 2019, and then even-odd-layer-dependent ferromagnetism in 2D CrCuSe_2 was confirmed by Wu et al in 2023. In this manuscript, the authors provide solid and complete experimental evidence that multiferroicity indeed exists in 2D CuCrSe_2 . I think this work deserves to be published in NC after addressing the following minor issues.

1. Please provide the preparation method and photographs of CuCrSe_2 single crystals for electrochemical exfoliation.
2. What is the magnetic anisotropy of 2D CuCrSe_2 material?
3. The magnetic measurement data of the samples should be further analyzed.
 - i. What is the value of effective magnetic moment per Cr ion.
 - ii. The large discrepancy between the ZFC and FC magnetization curves below about 75 K should be an interesting phenomenon, which may be ascribed to the micromagnetism caused by residual antiferromagnetic coupling in the sample. The discrepancy between the ZFC and FC curves reflects the pinning effect of antiferromagnetic coupling on ferromagnetic coupling.

Response to Reviewer #1

C1: *The manuscript by Sun et al. presents a significant discovery of coexisting high-temperature ferroelectricity and ferromagnetism in a single-layer, non-van der Waals material, CuCrSe₂. Employing a comprehensive suite of experimental techniques such as SHG, PFM, and STEM, the authors discovered ferroelectricity induced by Cu⁺ displacement. Additionally, they confirmed the coexistence of ferromagnetism through magnetic measurements and Hall effect experiments. The manuscript is well-structured, and the presented data are robust.*

I am particularly impressed by the high transition temperature, remarkable chemical stability, and the novel mechanism of multiferroicity in CuCrSe₂, where ferromagnetism is augmented by ferroelectric polarization. Given the growing interest in intrinsic 2D multiferroics, this manuscript addresses a highly relevant topic. Therefore, I recommend its publication in Nature Communications. However, before proceeding to publication, several issues should be addressed to further improve the manuscript.

Response:

Thank you for the careful and in-depth review of our manuscript. In the following, we will address all the comments and revise the manuscript accordingly.

C2: *The manuscript labels the atomic structure in Figure 1a with “A, B, C” but fails to provide a corresponding description in the main text.*

Response:

Thank you for the comment. To address this discrepancy, we added a sentence “as denoted by the labels A, B, and C” following “with alternating layers of CrSe₂ and Cu atoms stacked along the c-axis” in the revised manuscript (second paragraph, page 4).

C3: *The authors should compare the ferroelectricity in CuCrSe₂ from that observed in other 2D ferroelectrics, such as WTe₂, In₂Se₃, and CuInP₂S₆, to highlight the unique aspects of their discovery.*

Response:

Thank you for the suggestion. On the one hand, the coercive voltage of single-layer CuCrSe₂ (4-5 V) surpasses that of many conventional 2D ferroelectrics, a fact already highlighted in the main text (first paragraph, page 6). On the other hand, most 2D ferroelectrics are layered van der Waals materials, potentially limiting the diversity of available species. In this context, the investigation of room-temperature ferroelectricity in single-layer CuCrSe₂ represents a significant expansion of the ferroelectric domain to include non-layered 2D materials. To address this point, we added a sentence “In addition, compared with conventional layered van der Waals 2D ferroelectrics, the emergence of room-temperature ferroelectricity in non-layered CuCrSe₂ broadens the spectrum of ferroelectric materials” at the end of first paragraph, page 6.

C4: *Figures 3a and 3c lack scale bars, an essential detail for interpreting the images accurately.*

Response:

Thank you for the comment. Following your suggestion, we have incorporated a

scale bar into both Figure 3a and Figure 3c, as depicted in Figure R1.

Figure R1. Cross-section high-resolution STEM images of the two ferroelectric CuCrSe₂ phases along the [100] zone axis, P1 and P2. Scale bar: 0.5 nm.

C5: *In discussing in-plane ferroelectricity in monolayer CuCrSe₂, the manuscript references hysteresis in a two-terminal memristor device. To demonstrate more effectively high- and low-resistance switching, it would be beneficial if the authors could present I-V curves obtained under varying poling voltages.*

Response:

We appreciate your insightful suggestion regarding the in-plane ferroelectricity. Upon further examination of the crystal structure, prompted by Reviewer #2's remarks, we have concluded that in-plane ferroelectricity should not exist in CuCrSe₂. This determination is based on the fact that both polarization states, P1 and P2, are associated with the R3m space group, which features a mirror plane, *i.e.*, the (110) plane, that precludes in-plane polarization. Consequently, we have removed all references to in-plane ferroelectricity from the original manuscript, thereby clarifying that only out-of-plane ferroelectricity is feasible in CuCrSe₂.

C6: *References to “Extended Data Fig. X” within the main text should be corrected to “Supplementary Fig. X” to ensure consistency and clarity in document navigation.*

Response:

Thank you for your observation. We have amended all instances of the “Extended Data Fig. X” to “Supplementary Fig. X”.

C7: *Figure 4a showcases temperature-dependent magnetization measurements of CuCrSe₂ nanosheets, which primarily consist of ultrathin layers of CuCrSe₂. To provide a comprehensive understanding of the material's ferromagnetic properties, it is advisable for the authors to perform magnetization measurements on bulk CuCrSe₂.*

Response:

Thank you for your constructive comment. To elucidate the ferromagnetic properties of ultrathin CuCrSe₂ layers more effectively, we performed magnetization measurements on bulk CuCrSe₂, with the result displayed in Figure R2. It is clear from these findings that upon decreasing the temperature, bulk CuCrSe₂ exhibits characteristic antiferromagnetic behavior with a Neel temperature of 55 K, aligning with findings previously reported in the literature [Mater. Chem. Phys. 145, 156 (2014)].

Accordingly, we have included Figure R2 in the revised supplementary information as Figure S5.

Figure R2. M-T curves of CuCrSe₂ bulk crystal.

C8: *A recent study published in Nature Communications (Vol. 15, p. 721, 2024) on a one-unit-cell thick, interfacially modulated multiferroic Cr₂S₃ should be discussed. Although it explores a different mechanism—relying on interfacial interactions for multiferroicity, unlike the intrinsic high-temperature multiferroicity reported here—it is pertinent for contextualizing the current work within the broader research landscape.*

Response:

Thank you for the suggested reference. Following your suggestion, we have cited the paper in our revised manuscript as Ref. 32. Additionally, we have included a brief discussion at the end of the second paragraph on page 3, which reads: “Another significant advancement in the pursuit of single-layer multiferroicity is the fabrication of one-unit-cell-thick multiferroic Cr₂S₃; however, its 2D ferroelectric properties are attributed to interfacial modulation. Hence, the quest for a single material that exhibits intrinsic 2D multiferroicity remains a formidable challenge”.

Response to Reviewer #2

C1: *This manuscript reports the multiferroic behavior of thin flake samples of CuCrSe₂ obtained by redox-controlled chemical exfoliation. Two-dimensional materials are one of the hottest topics in the condensed matter physics nowadays, and the development of multiferroic two-dimensional materials will attract many readers. Breaking the inversion symmetry and ferromagnetism are clearly demonstrated in the manuscript. On the other hand, I am not convinced that all the experimental data are correctly interpreted. I also wonder if the information about the methodology is well provided. I hence have several questions and comments. I may recommend the publication of this work after the authors address all the following issues.*

Response:

We express our sincere gratitude for your review and constructive comments on our manuscript. These insights and suggestions have significantly enhanced the quality of our work. Below, we address each of your comments in sequence.

C2: Question 1. Is the chemical formula of the film really CuCrSe_2 ?

The authors first say, “The single-layer CuCrSe_2 consists of two CrSe_2 layers and one Cu layer”, which literally means the composition of $\text{Cu}_1\text{Cr}_2\text{Se}_4$. If this is the case, I recommend that the authors should refer to the material as CuCr_2Se_4 . Then, the Cu ion likely behaves as a divalent ion of $S=1/2$. This kind of valence change may happen since the samples were obtained by redox-controlled chemical exfoliation. The change in Cu valence can provide an enormous impact on the magnetism of the material.

Response:

Thank you very much for the insightful comments. We concur that single-layer CuCrSe_2 should more accurately be denoted as $\text{Cu}_1\text{Cr}_2\text{Se}_4$, and bilayer CuCrSe_2 as $\text{Cu}_2\text{Cr}_3\text{Se}_6$. Given that CuCrSe_2 is the chemical formula for the bulk crystal, adopting distinct compositional formulas for few-layered crystals might indeed lead to confusion among readers. Therefore, we have chosen to employ “1L CuCrSe_2 ” and “2L CuCrSe_2 ” to denote these ultrathin flakes. This approach aligns with designations used in previous literature [Nat. Chem. 13, 1235 (2021); Adv. Mater. 35, 2209365 (2023)].

To better clarify this point, following your suggestion, we added the description “Although the chemical formula should more accurately be denoted as CuCr_2Se_4 , we continue to use “1L CuCrSe_2 ” for simplicity and to avoid confusion, given that CuCrSe_2 is the chemical formula of the bulk crystal.” after “covalent bonded to one another, as shown in Fig. 1b” (second paragraph, page 4).

For single-layer CuCrSe_2 (or CuCr_2Se_4), we totally agree with you that the Cu ion behaves as a divalent ion. However, according to theoretical calculations, the Cu atoms do not contribute directly to the magnetic moments, as shown in Fig. R3. In our manuscript, we have indicated that the vertical electric polarization induced by the Cu atoms can enhance the ferromagnetic coupling. However, the potential impact of a change in Cu valence on the magnetism of the material requires further investigation.

[Redacted]

Figure R3. The cross-view structure for single-layer CuCrSe_2 and corresponding atomic magnetic moments, cited from Adv. Mater. 35, 2209365, 2023 (Supporting Information: Figure S8).

C3: Question 2: Is the sample really of single layer?

According to the previous report on the crystal structure of bulk CuCrSe_2 , the lattice parameter c in the hexagonal setting is approximately 1.94 nm (Gagor et al., Materials Chemistry and Physics 146, 283 (2014)). Since the hexagonal unit cell contains three layers, the thickness of single layer is estimated to be 0.6 or 0.7 nm. The obtained step of 1.6 nm is much larger than the thickness.

Response:

Thank you for this comment. As discussed in the response to C2 and in line with

conventions found in the literature, a single-layer CuCrSe_2 is composed of two CrSe_2 layers and one intervening Cu layer. This configuration results in a thickness greater than that of the lattice parameter c , where a unit cell is defined by one CrSe_2 layer and one Cu layer. Given that the thickness of a single CrSe_2 layer is approximately 0.6 to 0.7 nm, the theoretical thickness for 1L CuCrSe_2 is estimated to be about 1.4 to 1.6 nm. This estimation aligns closely with our experimental finding of 1.6 nm.

C4: Question 3: How was the 2L and 4L samples fabricated and characterized?

There is no description about the fabrication and characterization of the bilayer sample used in the PFM measurement and 2L and 4L samples used in the Hall resistivity measurement.

Response:

Thank you for the comment. All few-layered CuCrSe_2 flakes employed in the PFM and Hall measurements were fabricated by the same redox-controlled chemical exfoliation method. Typically, this process yields a large quantity of nanoflakes comprising various layers. Samples with 1 layer (1L), 2 layers (2L), and 4 layers (4L) were identified and selected based on AFM verification. Figure R4 showcases the morphology images of the 2L and 4L flakes utilized in our experiments, with their thicknesses determined to be approximately 2.3 nm and 3.6 nm, respectively.

Following your suggestion, we added Figure R4 to the revised supplementary information as Figure S7. We also added a sentence “*Thin flakes with different thicknesses were selected by atomic force microscopy measurements, as shown in Supplementary Fig. 7*” to the Methods section.

Figure R4. AFM image and height profile along the dashed line for 2L and 4L CuCrSe_2 sample.

C5: Question 4: In what configuration was the second harmonic generation measured? Fig. 2a just shows that the second harmonic is generated. However, the result does not tell the presence of electric polarization because of the lacking in the information on the experimental configuration. If the incident direction of the excitation laser was normal to the sheet, Fig. 2b shows the xxx , xyy , xyx , and yyy components of the second harmonic generation tensor. The azimuthal angle dependence shows threefold

symmetry and inversion breaking but not the out-of-plane polarization. For example, P-6 system with no polarization may also show a similar result. In addition, the authors should explicitly show the angles corresponding to the a and b axes of the crystal.

Response:

Thank you for the insightful question. The SHG measurements were conducted using a backscattering geometry, implying that the excitation laser's incidence direction was perpendicular to the sample surface. To clarify the experimental setup, we added the phrase “*based on the backscattering geometry*” to the original sentence “*we employed a 1064 nm picosecond laser as the excitation source and observed a robust emission peak centered at 532 nm*” (second paragraph, page 5).

We totally agree with your assessment that “*the azimuthal angle dependence shows threefold symmetry and inversion breaking but not the out-of-plane polarization*”. Consequently, in our manuscript, SHG has been utilized solely to ascertain structural asymmetry. The presence of out-of-plane polarization was further substantiated through a comprehensive series of PFM measurements.

Following your valuable suggestion, we have added an inset to Figure 2a to illustrate the relative angle (θ) in the azimuthal angle-dependent SHG results (Figure 2b), as depicted in Figure R5.

Figure R5. Optical image of the sample used for SHG measurements, labeled with the initial polarization of the laser and crystalline orientation, the rotation angle is denoted as θ . Scale bar: 20 μm .

C6: *Question 5: How thick is the nanosheet used for the magnetization measurement demonstrated in Fig. 4?*

Response:

Thank you for your question. The magnetization measurements depicted in Figures 4a and 4b involved a composite of numerous ultrathin CuCrSe_2 nanosheets, with an average thickness of fewer than 8 layers, as shown by Figure R6. To depict the ferromagnetic characteristics of single-layer CuCrSe_2 more accurately and various few-layered CuCrSe_2 crystals, we carried out Hall measurements on these distinct layers. The results of these investigations are presented in Figure 4d within the main text.

We added Figure R6 to the revised supplementary information as Figure S6.

Figure R6. Typical optical image of the chemical exfoliated CuCrSe₂ nanosheets used for the magnetization measurement.

C7: Comment 1: *The authors claim that CuCrSe₂ also hosts the in-plane polarization, based on the double-well model shown in Fig. 3b. However, both of P1 and P2 states have the symmetry of R3m, at least in the bulk case. In other words, (110), (2 -1 0), and (-1 2 0) planes are reflection mirrors, which prohibits the in-plane polarization. Cu ion can move to one of the three adjacent unoccupied tetrahedral sites. Even if all the Cu ions move in the same direction at the same time, this should be regarded as ionic conduction, which has intensively been discussed since 1980's.*

Response:

Thank you for your insightful comment. We apologize for our oversight in not carefully examining the crystal structure. As you correctly pointed out, and as depicted in Figure R7, a mirror plane exists in CuCrSe₂ irrespective of whether it is in the P1 or P2 state, which precludes in-plane polarization. Given this structural characteristic, we have excised all descriptions of in-plane ferroelectricity from our manuscript, affirming that only out-of-plane ferroelectricity exist in CuCrSe₂.

While we agree with the possible ionic conduction within this system, as you pointed out, experimental confirmation requires additional efforts. We hope that our findings will encourage further investigations into the possible interplay among the ionic conduction, ferroelectricity, and ferromagnetism. In our current study, the possible ionic conduction does not impede our claim of multiferroicity in CuCrSe₂. Therefore, we have opted not to delve into this aspect within the present manuscript.

Figure R7. Top view of the in-plane crystal structure of CuCrSe₂. (110) is one of the reflection mirror planes.

C8: Comment 2: *The order-disorder type structural phase transition in CuCrSe₂ was*

reported in 2014 by Gagor and collaborators (Materials Chemistry and Physics 146, 283 (2014)). The paper also reported the ordering of the tetrahedral site occupation was essential for the transition. I believe that this paper has to be cited.

Response:

Thank you for recommending this interesting paper. This paper reported an order-disorder phase transition ($T=365$ K) in CuCrSe_2 crystal. Below this transition temperature, which includes room temperature, an imbalanced occupation of interlayer Cu atom sites occurs, giving rise to natural ferroelectric polarization. We have cited this study in the revised manuscript as Ref. 48.

C9: *Comment 3: Y. Zhou et al. very recently reported the ferroelectric thin film of CuCrS_2 (Nano Letters 24, 2118, (2024)), which should be added as a reference.*

Response:

Thank you for recommending this interesting paper. We have cited the paper in the revised manuscript as Ref. 27.

C10: *Comment 4: Can the authors explain why the Hall resistance is smaller in a thinner flake? It is counterintuitive.*

Response:

Thank you for the comment. After the Hall data acquisition, we also got a little bit confused about the Hall resistance. Especially for the single-layer CuCrSe_2 sample, the R_{xy} value is significantly lower than that of the bilayer sample. A plausible explanation for this discrepancy could lie in the variance in device details, including differences in metal-surface contact and sample quality. Similar counterintuitive results have been reported in several articles, such as in Nat. Commun. 11, 3729 (2020) and Nat. Mater. 20, 818 (2021). In these studies on layered 2D ferromagnets (FeTe and CrSe_2), it was noted that the Hall resistance was also smaller in thinner flakes, although the authors did not elaborate on the underlying reasons for this phenomenon.

Response to Reviewer #3

C1: *2D multiferroic materials have garnered broad interests attributed to their magnetoelectric properties and multifunctional applications. The simultaneous coexistence of robust magnetism/ferroelectricity in ultrathin-layer CuCrSe_2 was previously predicted via first-principles calculations by Liu et al in 2019, and then even-odd-layer-dependent ferromagnetism in 2D CrCuSe_2 was confirmed by Wu et al in 2023. In this manuscript, the authors provide solid and complete experimental evidence that multiferroicity indeed exists in 2D CuCrSe_2 . I think this work deserves to be published in NC after addressing the following minor issues.*

Response:

Thank you for the high evaluation of our work. In the following, we will address all the comments sequentially.

C2: Please provide the preparation method and photographs of CuCrSe₂ single crystals for electrochemical exfoliation.

Response:

Thank you for your comment. We have added the preparation method of CuCrSe₂ single crystal in the Methods part of the main text, which reads:

“Growth of CuCrSe₂ single crystals. Single crystals of CuCrSe₂ were synthesized using a chemical vapor transport (CVT) method starting from polycrystalline powders. Polycrystalline CuCrSe₂ was prepared by a solid-state reaction in which a stoichiometric mixture of high purity Cu, Cr, and Se powders was sealed in an evacuated quartz tube and sintered at 1050 °C for 40 h. The furnace was then heated to 1200 °C and kept for several minutes, followed with rapid quenching to obtain pure phase of CuCrSe₂. The polycrystalline CuCrSe₂ was then grinded into fine powder and sealed in an evacuated quartz tube together with a small amount of CrCl₃ as a transport agent. The tube was heated on a two-zone tube furnace in a temperature gradient of 1050 °C at the hot end and 800 °C at the cold end for 200 hours. Rapid quenching with cold water was also employed and single crystals of CuCrSe₂ were obtained” (page 9).

Figure R8 shows a photograph of a typical single crystal used for electrochemical exfoliation, and this picture has been added to the revised supplementary information as an inset in Figure S1.

Figure R8. Photograph of typical CuCrSe₂ single crystals.

C3: What is the magnetic anisotropy of 2D CuCrSe₂ material?

Response:

Thank you for the comment. The magnetic easy axis is along in-plane direction, which has been reported in a recent work by some authors of the current paper [Adv. Mater. 35, 2209365 (2023)].

The result is displayed in Figure R9. With the decrease of θ_H (the tilt angle between the magnetic field and basal plane of the CuCrSe₂ nanosheet) from 90° to 0°, R_{xy} gradually decreases. The tilt angle between the magnetization and the basal plane can be calculated using the formula,

$$\theta_M(\theta_H) = \sin^{-1} \left[\frac{R_{xy}(\theta_H)}{R_{xy}(\theta_H = 90^\circ)} \right]$$

Because θ_M is always smaller than θ_H (Figure R9c), there exists an in-plane magnetic anisotropy in single-layer CuCrSe₂.

To address this comment, we added a phrase “with a strong in-plane magnetic anisotropy” after “indicating that the magnetic order can persist down to monolayer”

(second paragraph, page 8).

[Redacted]

Figure R9. Angle-dependent Hall measurements of single-layer CuCrSe₂ at 2 K, cited from *Adv. Mater.* 35, 2209365, 2023 (*Supporting Information: Figure S5*).

C4: *The magnetic measurement data of the samples should be further analyzed.*

i. What is the value of effective magnetic moment per Cr ion.

ii. The large discrepancy between the ZFC and FC magnetization curves below about 75 K should be an interesting phenomenon, which may be ascribed to the mictomagnetism caused by residual antiferromagnetic coupling in the sample. The discrepancy between the ZFC and FC curves reflects the pinning effect of antiferromagnetic coupling on ferromagnetic coupling.

Response:

Thank you for these constructive comments.

- 1) Based on theoretical analyses [Adv. Mater. 35, 2209365 (2023)], within a single-layer CuCrSe₂, the spins of the Cr ions in the two constituent layers align parallelly, exhibiting magnetic moments of 2.83 and 2.96 μ_B , respectively, as shown in Figure R10. Experimentally determining the effective magnetic moment per Cr ion from our M-H curve (Figure 4b) is difficult. This difficulty arises because the sample used for M-H measurements consists of a multitude of ultrathin CuCrSe₂ nanosheets, where the effective magnetic moment is not constant but varies across different layers. Notably, as the number of layers increases, the magnetic moment tends to decrease. From our existing dataset, the estimated average effective magnetic moment for this collective sample is approximately 1.50 μ_B .

[Redacted]

Figure R10. The cross-view structure for single-layer CuCrSe₂ and corresponding atomic magnetic moments, cited from *Adv. Mater.* 35, 2209365, 2023 (*Supporting Information: Figure S8*).

- 2) Thank you for providing great insights to better understand the large discrepancy between the ZFC and FC magnetization curves below ~75 K. It is highly possible that the moment reduction at low temperature without external magnetic field is due to the residual antiferromagnetic coupling in the sample. Following your nice suggestion, we added a brief description “*Meanwhile, such a phenomenon might be ascribed to the mictomagnetism caused by residual antiferromagnetic coupling in the sample*” after “*This feature can be attributed to the fact that the applied external magnetic field is less than the material’s coercive field*” in the revised manuscript (first paragraph, page 8).

Summary of Changes

1. All changes in the main text are highlighted in the document “Manuscript (Tracked Version).pdf”.
2. Figure 2 and Figure 3 have been updated.
3. We added three new figures to the Supporting Information: Fig. S5, S6, and S8. Fig. S5 in the original version was deleted.

REVIEWER COMMENTS

Reviewer #1 (Remarks to the Author):

The authors have addressed all my comments and this manuscript can be published in Nat. Commun.

Reviewer #2 (Remarks to the Author):

I have investigated the authors' response and the revised manuscript. Most of the comments raised by the three referees have been satisfactorily addressed. However, I believe that some revisions are still necessary before the publication.

1. The authors add a phrase "as denoted by the labels A, B, and C" on line 89 to respond to the first comment of reviewer #1. As the labels are displayed in Fig. 1a, this figure number should be cited in the sentence.
2. The authors add a sentence "In addition, compared with conventional layered van der Waals 2D ferroelectrics, the emergence of room-temperature ferroelectricity in non-layered CuCrSe₂ broadens the spectrum of ferroelectric materials" on line 153 to respond to the second comment of reviewer #1. The phrase 'non-layered CuCrSe₂' sounds strange to me. For example, the title is "Evidence for Multiferroicity in Single-Layer CuCrSe₂".
3. I asked if the sample is really of single layer in the previous round. It should be noted that the conventional hexagonal unit cell of bulk CuCrSe₂ of space group R3m contains THREE Cu sheets and THREE CrSe₂ layers contrary to the authors' response, "where a unit cell is defined by one CrSe₂ layer and one Cu layer". The c parameter of bulk CuCrSe₂ is approximately 1.94 nm (Gagor et al., Materials Chemistry and Physics 146, 283 (2014)), showing that the thickness of two CrSe₂ layers and two Cu sheets is approximately 1.3 nm. The obtained step of 1.6 nm, larger than 1.3 nm, might correspond to the 2L CuCr₂Se₂ composed of two Cu sheets and three CrSe₂ layers.
4. I am happy to learn that the revised manuscript clearly mentions that the SHG measurement was performed in the back scattering geometry. On the other hand, I doubt if the inset added to Fig. 2a clearly tells the definition of theta. I do not understand what the black arrow, black dashed line, and white dashed line indicate, respectively. What is the theta value for the laser polarization parallel to the [100] axis in Fig. 2b? It is essential for confirming the agreement of the observed azimuthal dependence with the 3m structure model.
5. The authors add a sentence "Meanwhile, such a phenomenon might be ascribed to the mictomagnetism caused by residual antiferromagnetic coupling in the sample" on line 207 to respond to the third comment of reviewer #3. However, mictomagnetism is, in my opinion, a sort of spin glass in a broad sense, which might not appear in the present pure system.

Reviewer #3 (Remarks to the Author):

Issues raised in my review were properly addressed. I suggest to accept this paper for

publication.

Response to Reviewer #1

C1: *The authors have addressed all my comments and this manuscript can be published in Nat. Commun.*

Response:

We appreciate again for your thoughtful comments and nice recommendation, which greatly improve the readability and clarity of our manuscript.

Response to Reviewer #2

C1: *I have investigated the authors' response and the revised manuscript. Most of the comments raised by the three referees have been satisfactorily addressed. However, I believe that some revisions are still necessary before the publication.*

Response:

We express our sincere gratitude for your review and constructive comments on our revised manuscript. In the following, we will address all the concerns and revise the manuscript accordingly.

C2: *The authors add a phrase "as denoted by the labels A, B, and C" on line 89 to respond to the first comment of reviewer #1. As the labels are displayed in Fig. 1a, this figure number should be cited in the sentence.*

Response:

Thank you for the comment. We have changed the original sentence "as denoted by the labels A, B, and C" to "as denoted by the labels A, B, and C in Fig. 1a".

C3: *The authors add a sentence "In addition, compared with conventional layered van der Waals 2D ferroelectrics, the emergence of room-temperature ferroelectricity in non-layered CuCrSe₂ broadens the spectrum of ferroelectric materials" on line 153 to respond to the second comment of reviewer #1. The phrase 'non-layered CuCrSe₂' sounds strange to me. For example, the title is "Evidence for Multiferroicity in Single-Layer CuCrSe₂".*

Response:

Thank you for the comment. Here, the term "non-layered" is used to describe bulk crystals that are not of the van der Waals type, which would otherwise facilitate easy mechanical exfoliation into ultrathin flakes. This distinction underscores the significance of our chemical exfoliation technique for such materials.

According to your suggestion and drawing from recent literature on the chemical exfoliation of a similar compound, AgCrS₂ [Nat. Chem. 13, 1235 (2021)], we recognized that the term "non van der Waals" more accurately conveys the nature of CuCrSe₂. Consequently, we have revised the term "non-layered" to "non van der Waals" in the revised manuscript.

C4: *I asked if the sample is really of single layer in the previous round. It should be noted that the conventional hexagonal unit cell of bulk CuCrSe₂ of space group R3m contains THREE Cu sheets and THREE CrSe₂ layers contrary to the authors' response, "where a unit cell is defined by one CrSe₂ layer and one Cu layer". The c parameter of bulk CuCrSe₂ is approximately 1.94 nm (Gagor et al., Materials Chemistry and Physics 146, 283 (2014)), showing that the thickness of two CrSe₂ layers and two Cu sheets is approximately 1.3 nm. The obtained step of 1.6 nm, larger than 1.3 nm, might correspond to the 2L CuCr₂Se₂ composed of two Cu sheets and three CrSe₂ layers.*

Response:

Thank you for this insightful question. We apologize for our oversight in our previous response regarding the composition of a unit cell in bulk CuCrSe₂, which indeed comprises three Cu sheets and three CrSe₂ layers. Consequently, our description of single-layer CuCrSe₂ represents a structure thinner than one unit cell. Additionally, we recognized that the measured step height of 1.6 nm exceeds the estimated thickness of 1.3 nm.

It is well-established that AFM measurements can overestimate the step heights of 2D materials. This discrepancy is attributed to the distinct interactions between the AFM tip with the 2D materials and the substrate. Moreover, the presence of interfacial adsorbates can further inflate the measured step height.

Such overestimations are quite common and have been reported across various studies. For example, the measured thickness of bilayer CuInP₂S₆ was reported as 1.6 nm, surpassing the theoretical value of 1.35 nm [Nat. Commun. 7, 12357 (2016)]; similarly, the thickness of 1L-CrSBr was found to be approximately 1.4 nm, exceeding the theoretical value of 0.8 nm [Nat. Mater. 21, 754 (2022), see Fig. S3]. Graphene has also been subject to similar measurement discrepancies [Rev. Sci. Instrum. 90, 103702 (2019); Phys. Status Solidi C 7, 1251-1255 (2010)].

Indeed, the accurate determination of the thickness of ultrathin CuCrSe₂ flakes has previously been reported by some co-authors of this work [Adv. Mater. 35, 2209365 (2023)]. Utilizing the same methodology to fabricate ultrathin CuCrSe₂ flakes ensures the precise assessment of their thickness in this work.

Following your comment, we changed the sentence "*This measured thickness is consistent with the theoretical value for the unit cell of CuCrSe₂*" to "*This measured thickness corresponds to 1L CuCrSe₂, which is slightly larger than the theoretical value of ~1.3 nm. Similar overestimations have been reported for many other 2D materials [39-40]*".

To give the readers a better understanding of the relationship between the bulk and single-layer CuCrSe₂, we also revised the sentence to "*Although the unit cell of bulk CuCrSe₂ comprises three Cu sheets and three CrSe₂ layers, we define single-layer (1L) CuCrSe₂ as consisting of two CrSe₂ layers and one Cu layer, which are strongly covalently bonded to each other, as illustrated in Fig. 1b. The addition of one more CuCrSe₂ layer includes one more CrSe₂ and Cu layers*" in the second paragraph of Page 4. This definition is consistent with previous literatures on similar compounds.

C5: *I am happy to learn that the revised manuscript clearly mentions that the SHG measurement was performed in the back scattering geometry. On the other hand, I*

doubt if the inset added to Fig. 2a clearly tells the definition of theta. I do not understand what the black arrow, black dashed line, and white dashed line indicate, respectively. What is the theta value for the laser polarization parallel to the [100] axis in Fig. 2b? It is essential for confirming the agreement of the observed azimuthal dependence with the 3m structure model.

Response:

Thank you for raising this point. We apologize for the misunderstanding caused by our previous description.

The white dashed line in the optical image indicates the [100] axis of CuCrSe₂, with the sample position remaining fixed throughout the polarization dependent SHG measurement. The black solid arrow indicates the initial polarization direction of the incident laser ($\theta=0^\circ$).

To enhance the presentation of the data, we implemented an offset of approximately 20° in Figure 2b of the main text, thereby achieving a greater symmetry. Consequently, the initial polarization direction of the incident laser is parallel to the [100] axis. Then, θ in Figure 2b will be defined as the relative angle between the polarization direction and the [100] axis, as illustrated in Figure R1.

In the revised manuscript, we have replaced the inset of Figure 2a with Figure R1. Furthermore, we have revised the caption to provide readers with a clearer comprehension of the measurement conditions.

Figure R1. The corresponding optical image showcases the sample, with the angle θ defined as the relative angle between the [100] axis (indicated by a white dashed line) and the direction of laser polarization. Scale bar: 20 μm .

C6: *The authors add a sentence “Meanwhile, such a phenomenon might be ascribed to the mictomagnetism caused by residual antiferromagnetic coupling in the sample” on line 207 to respond to the third comment of reviewer #3. However, mictomagnetism is, in my opinion, a sort of spin glass in a broad sense, which might not appear in the present pure system.*

Response:

Thank you for your insightful comment. We conducted further literature review and found that this concept aligns more closely with ferromagnetic alloys or glasses. Just as you suggested, mictomagnetism is likely not applicable to the current system. Consequently, we have revised our statement to “Meanwhile, such a phenomenon might

be ascribed to the residual antiferromagnetic coupling in the sample". We hope that our work can stimulate further research on this material.

Response to Reviewer #3

C1: *Issues raised in my review were properly addressed. I suggest to accept this paper for publication.*

Response:

We appreciate again for your thoughtful comments and nice recommendation, which greatly improve the readability and clarity of our manuscript.

Summary of Changes

1. All changes in the main text are highlighted in the document "Revised Manuscript (Tracked Version).pdf".
2. The inset of Fig. 2a have been replaced.

REVIEWERS' COMMENTS

Reviewer #2 (Remarks to the Author):

I am pleased to find that all the comments are addressed in a satisfactory way. I now recommend that the paper be published in Nature Communications.